# Natural Polyphenol Recovery from Apple-, Cereal-, and Tomato-Processing By-Products and Related Health-Promoting Properties

**DOI:** 10.3390/molecules27227977

**Published:** 2022-11-17

**Authors:** Katalin Szabo, Laura Mitrea, Lavinia Florina Călinoiu, Bernadette-Emőke Teleky, Gheorghe Adrian Martău, Diana Plamada, Mihaela Stefana Pascuta, Silvia-Amalia Nemeş, Rodica-Anita Varvara, Dan Cristian Vodnar

**Affiliations:** 1Institute of Life Sciences, University of Agricultural Sciences and Veterinary Medicine, 400372 Cluj-Napoca, Romania; 2Department of Food Science and Technology, University of Agricultural Sciences and Veterinary Medicine, 400372 Cluj-Napoca, Romania

**Keywords:** agro-industrial by-products, biological activity, circular economy, health effects, re-integration, waste management, phenolic compounds

## Abstract

Polyphenols of plant origin are a broad family of secondary metabolites that range from basic phenolic acids to more complex compounds such as stilbenes, flavonoids, and tannins, all of which have several phenol units in their structure. Considerable health benefits, such as having prebiotic potential and cardio-protective and weight control effects, have been linked to diets based on polyphenol-enriched foods and plant-based products, indicating the potential role of these substances in the prevention or treatment of numerous pathologies. The most representative phenolic compounds in apple pomace are phloridzin, chlorogenic acid, and epicatechin, with major health implications in diabetes, cancer, and cardiovascular and neurocognitive diseases. The cereal byproducts are rich in flavonoids (cyanidin 3-glucoside) and phenolic acids (ferulic acid), all with significant results in reducing the incidence of noncommunicable diseases. Quercetin, naringenin, and rutin are the predominant phenolic molecules in tomato by-products, having important antioxidant and antimicrobial activities. The present understanding of the functionality of polyphenols in health outcomes, specifically, noncommunicable illnesses, is summarized in this review, focusing on the applicability of this evidence in three extensive agrifood industries (apple, cereal, and tomato processing). Moreover, the reintegration of by-products into the food chain via functional food products and personalized nutrition (e.g., 3D food printing) is detailed, supporting a novel direction to be explored within the circular economy concept.

## 1. Introduction

By-products of agro-industrial provenance and food waste are severe global concerns, particularly in many developed countries [1]. Food consumption has increased as a consequence of urbanization, population expansion, and economic growth, and it remains a consistent issue worldwide in the long run [2,3]. The most bothersome sectors are the wheat flour production, apple juice production, and tomato-processing industries, which generate massive amounts of residues, as a result of the extensive yearly processed tonnage (Figure 1). On the other hand, the low cost and straightforward availability of this residual biomass shelter the economic prospects of its potentially valuable components [4].

In general, cereal grains are the world’s main food source, contributing up to 300 million tons (Mt) of food products yearly according to FAOSTAT (http://www.fao.org/ accessed on 10 October 2022) [5]. The primary cereals produced are maize, wheat, rice, barley, and oat crops, from which maize is mainly submerged as flour, animal feed, and substrate in ethanol production [6], while wheat is consumed worldwide mostly in the form of flour [7].

Maize (*Zea mays* L.), cultivated globally, is the prime cereal grown in the world [8]. As presented in Figure 1A, its production increased by more than 50% in the last 20 years and is the most significant starch supply, and subsequently, it is one of the main organic by-product generators in the world [6]. In Europe, cereal production increased more than four times from 2000 to 2020, with the highest production in Western and Eastern Europe (Figure 1B). According to a recent study, in Europe, these cereal by-products, via integration with foodstuffs, could replace as many as 8.8 Mt of edible grains [9].

Wheat (*Triticum aestivum* L.) is one of the world’s most prominent cereals. According to FAOSTAT, the global output in 2020 was over 760 million tons (Mt), up from 580 Mt in 2000, showing a 30.97% increase in wheat production over the last two decades, and the global output is expected to increase gradually in the future [10]. In agreement with the European Flour Millers’ Report (2016), more than 45 Mt of wheat and oats is processed annually in Europe, creating over 6.5 Mt of by-products, which are predominantly used for animal nutrition [11]. By-products such as bran are a rich source of beneficial phytochemicals such as dietary fibers, minerals, vitamins, polyphenols, and phytosterols, known for their health-promoting properties [7,11,12,13]. Phenolic acids are among these components since they are categorized as bioactive phytochemicals and have a significant impact on human health [14].

Apple (*Malus domestica* sp.) is one of the most widespread fruits on the global scale. In 2020, the global output was over 86 Mt, up from just over 59 Mt in 2000, indicating an incremental tendency of apple production, with 46.19% in the last two decades [5]. China is the leading contributor to overall production with 46.85%, followed by the USA with 5.38%, and Turkey with 4.97%, and from Europe, the main contributor is Poland with 4.11%, as shown in Figure 1C. Furthermore, the worldwide output is predicted to rise steadily in the future [15,16].

The resulting by-product of apple pressing/processing, which derives from juice, cider, wine, distilled spirits, and vinegar production, as well as the formulation of jellies, is known as apple pomace (AP). Solid pomace makes up to 20–35% of the apple fruit’s fresh weight and is composed of a mixture of pulp, peel, core, seed, and calyx. Approximately 95.5% of the solid waste is produced by epi-mesocarp [17]. Dietary fiber, which makes up around 65% of the dry weight of AP, is the main component from a nutritional point of view, and the majority of dietary fiber in all pomace is insoluble. Apple seeds include significant amounts of proteins and lipids, up to 49.5 and 24%, respectively. Additionally, hemicellulose is the second-most significant fiber in AP (19.9–32.2%), and cellulose accounts for a crucial part, comprising 43% of the pomace [18]. The by-products of the seed and peel are rich in phenolic chemicals, primarily chlorogenic acid and phloridzin [19].

Tomatoes (*Solanum lycopersicum* L.) are considered one of the world’s most popular vegetables but are classified as a fruit. In Europe, it was first domesticated in Spain [20], and the main producers are represented in Figure 1D. The global output of tomatoes increased to 186 Mt in 2020, up from just over 109 Mt in 2000 [5]. According to FAOSTAT, a substantial increase in tomato production can be observed in the last two decades, precisely above 70%. The major producers are China with 34.67%, followed by India with 11.01%, Turkey with 7.07%, and the USA with 6.54% of global tomato crop production. Additionally, the global output is expected to increase gradually in the future [2]. More than one-third of the tomatoes are processed due to their perishable nature, and considerable amounts of by-products are generated via juice, sauces, ketchup, or puree production, among other meals. The removal of tomato peels, seeds, and tiny amounts of pulp can add between 5 and 30% to the cost of the primary products, and the by-products are a source of environmental discomfort due to their high moisture content, which favors bacterial growth and carbon dioxide emission [21]. The scientific literature confirms the presence of health-promoting compounds such as carotenoids, polyphenols, tocopherols, terpenes, and sterols in industrial tomatoes and their by-products, and endorses their extraction and revalorization in functional food products, as the bioactive substances resist to industrial processes [22,23].

Polyphenols are secondary metabolites that contain one or more hydroxyl groups, being one of the largest classes of valuable bioactive compounds for supporting human health [24,25]. Phenols are essentially made up of a hydroxyl group (-OH) linked directly to an aromatic hydrocarbon group, whereas polyphenols are larger polymers of 12 phenolic hydroxyl groups linked to five-to-seven aromatic rings [26,27]. Based on recent trends regarding healthier lifestyles and an increase in the consumption of foodstuffs derived from natural resources without any environmental drawbacks, the reintegration of by-products generated by the food industry is an important topic [28,29].

Green alternatives to lessen environmental pollution and waste generation include renewable biomaterials, and wheat-, apple-, and tomato-processing by-products can be framed into this category. In recent years, innovative and inventive applications of these by-products have contributed to the steady advancement of bioeconomy and biotechnology through the extraction and/or revalorization of natural phenolic compounds [30].

The use of agro-industrial derivatives might provide an additional source of income, and it could reduce by-product disposal and improve the nutritional profile of functional food items at the same time. Utilizing grain, tomato, and apple by-products from industrial production may provide a generous supply of nutrients, and their repurposing might represent a substantial source of revenue. Therefore, the objective of the current literature review is to highlight the primary phenolic components linked to human health, which can be found in the by-products of these three major agrifood industries (apple, cereal, and tomato processing), and to encourage their recovery and, accordingly, reintegration into the food chain using circular economy principles. Nonetheless, based on the findings of this study, it is projected that the food-processing industries could better manage their by-products and waste (e.g., through the incorporation of by-products into food formulations to boost the nutritional value), therefore avoiding a major environmental concern.

## 2. Polyphenols in Apple-Processing By-Products

Apples are one of the most consumed fruits worldwide, both in industry and at the individual population level [31]. Approximately 11 million metric tons of apples is produced and used annually in the apple-processing industry and alcoholic beverage production in Europe [32]. Apple pomace is one of the most widely produced agrifood wastes, with an annual production rate of about 4 million tons worldwide [33]. The recovery rate for this by-product, however, is modest. Pomace is frequently discarded and dumped in landfills as waste, which causes environmental issues and presents a potential risk to public health [3,34].

The amount of pomace resulting after apple processing can be reused in biotechnological routs as a substrate for the production of different compounds, such as flavoring compounds, pigments, fuel, and citric acid, or as raw materials for the extraction of fibers and phenolic compounds [35,36,37,38].

From a nutritional point of view, apple pomace is a by-product rich in fibers, vitamins, minerals, phenolic compounds, and pigments [19]. All these macronutrients have a significant role in the human organism through their effects on metabolism. Therefore, apple pomace has attracted researchers’ consideration, as well as stakeholders’ attention, by virtue of its valuable composition and by presenting suitable properties for further sustainable use [39].

The nutritional profile of apple pomace is mainly represented by phenolic compounds, carbohydrates, and fibers, as presented in table Table 1. These constituents can help treat gastrointestinal disorders, decrease serum triglycerides and LDL-cholesterol, and regulate glycemia [40,41]. All these effects in the human organism can be explained through their high concentration of the beneficial compounds mentioned above, primarily exerting anti-inflammatory and antioxidant roles [42].

In addition to the benefits, the consumption of apple pomace may raise questions associated with toxicity when it comes to its applicability in the food industry [44], with seeds representing 4–5% of the apple pomace [45]. Apple seeds contain a plant toxin called cyanogenic glycoside amygdalin, which can interact with digestive enzymes, resulting in the release of hydrogen cyanide. This toxin can cause different symptoms, from mild symptoms such as dizziness to severe symptoms such as paralysis and coma [46]. However, to reach poisoning levels, the ingested quantity must be significantly high; more exactly, between 83–500 apple seeds are needed to reach the poisoning level, or the person must consume more than 800 g of apple pomace [33,47]. Studies on apple seed oil have confirmed the safety of its use, as the limit was found to be below the toxicity level [48].

Phenolic compounds are concentrated mainly in the core, seeds, peel, calyx, and stem, as well as in smaller amounts in the pulp, highlighting how apple pomace can be valorized through its high amount of antioxidant compounds. As shown in Figure 2, the total phenolic content of seeds has a higher value compared with the pulp, followed by the peel, both being part of apple pomace.

The predominant phenolic compound families in apple pomace are dihydrochalcones, procyanidins, flavan-3-ol monomers, flavonols, anthocyanidins, and hydroxycinnamic acids. The most representative compounds are phlorizin from the dihydrochalcones family, chlorogenic acid from the hydroxycinnamic acids family, and epicatechin from the flavan-3-ol monomer family [50].

One of the representative phenolic compounds in apples and, remarkably, apple pomace, is phlorizin (Figure 3A). As the main compound from the dihydrochalcone family, the amount of phlorizin in apple pomace is approximately 1.6 mg/g dry weight, highlighted in a study by Lavelli et al. on the stability of phenolic compounds in apple pomace [51]. Phlorizin is used as a marker of apple varieties and is mainly concentrated in apple seeds. This polyphenol is also used as a reliable marker for spotting the presence of apples, a less expensive alternative compared with the reported fruits [52].

Nevertheless, it also acts as a strong antioxidant, anti-inflammatory, and antimicrobial compound [53,54]. Regarding its benefits, phlorizin exerts several health benefits, mainly in diabetes, due to its ability to alter the glucose absorbed and excreted. A recent study illustrated that the intestine and kidney’s sodium/glucose cotransporters are specifically and competitively inhibited by phlorizin. In addition, postprandial hyperglycemia therapy in diabetes and other associated illnesses, such as obesity, may benefit from this characteristic [55]. A study conducted on streptozotocin-induced diabetic mice showed that a diet containing 0.5% phlorizin significantly improves the exacerbated elevations in blood glucose levels [56,57]. Another health benefit can be seen in colitis, where it acts as a protective compound for the intestinal brush border [58].

Chlorogenic acid (Figure 3B) is representative in the peel and flesh of apples compared with their seeds. Chlorogenic acid is a powerful antioxidant known for counteracting pathologies caused by oxidative processes [59,60]. A study conducted on the improvement of mood and cognition in the elderly showed enhanced results after the administration of enriched decaffeinated coffee with chlorogenic acid, displaying that the consumption of products containing chlorogenic acid can help in the treatment of neuro-cognitive diseases [61]. Besides the benefits mentioned previously, chlorogenic acid can also confer positive effects by lowering blood pressure, confirmed in a randomized trial involving healthy volunteers after the administration of 400 mg chlorogenic acid in 400 mL low-nitrate water. This effect can be explained by the ability of phenolic compounds to increase nitric oxide, which improves cardiovascular health [62].

The third phenolic compound found in apple pomace in smaller amounts is epicatechin (Figure 3C). Besides all its fulfilled roles (e.g., antioxidation, anti-inflammation), epicatechin can exert its role in diabetes, cancer, and cardiovascular disease, acting as a neuroprotective compound, and it improves muscle performance [63]. Cilleros et al. showed, in an in vitro study, the effect of epicatechin in regulating glucose uptake and bolstering the insulin signaling pathway [64]. A study conducted on human monocytic cells (THP1 cells) showed similar results regarding the beneficial effects of epicatechin in diabetes through the attenuation of high-glucose-induced inflammation [65].

## 3. Polyphenols in Cereal-Processing By-Products

Rice, wheat, barley, and corn are the most prevalent cereals globally, accounting for more than 90% of total cereal consumption [8]. By-products from the grain processing food industry include bran, straw, germ, and spent brewery grain [8,14]. Recent scientific studies confirm that the by-products of grain milling contain a wide range of beneficial components that support human health. These components, illustrated in Figure 4, include dietary fiber, vitamins, minerals, phytosterols, and polyphenols [66].

The most common polyphenols in cereal by-products are flavonoids and phenolic acids, categorized into two large classes: hydroxybenzoic acids and hydroxycinnamic acids [67]. Gallic, vanillic, p-hydroxybenzoic, protocatechuic, syringic, and salicylic acids are all hydroxybenzoic acids with a C6-C1 structure. Hydroxycinnamic acids, on the other hand, have a C6-C3 structure comprising caffeic, ferulic, p-coumaric, chlorogenic, and synaptic acids [68].

Most of the phytochemicals found in cereal grains are identified in the bran fraction and are primarily bound to dietary fiber components, such as cellulose, hemicellulose, and lignin. In cereal by-products, a smaller amount of phenolic acid is present in a free form, and a larger amount is present in a bound form. In corn, for example, the free phenolic fraction ranges between 1 and 5 mg/100 g DW, while the bound fraction ranges between 150 and 300 mg/100 g DW [69]. The common phytochemical composition of cereal bran is presented in Table 2.

Ferulic acid is primarily found in cereal by-products with values over 1000 μg/g DW [69]. Wheat bran contains 90% ferulic acid, while oat bran contains 75% ferulic acid [84]. The metabolism of phenylalanine and tyrosine is responsible for their widespread presence in plant-based sources. Ferulic acid may be found in cereals in several forms, including free ferulic acid, γ-oryzanol, ferulic acid monoesters, and certain triterpene alcohols [84]. Ferulic acid is covalently linked to cell wall molecules, such as lignin, polysaccharides, long-chain fatty acids, and flavonoids via ester, ether, and amide bonds [85,86]. Because of the position of the hydrogen atom in the hydroxyl group of ferulic acid, it reacts with a free radical to produce an antioxidant effect [69,87].

Cereal bran is also rich in vanillic, syringic, salicylic, caffeic, and p-coumaric acids [68]. Vanillic acid (4-hydroxy-3-methoxybenzoic acid) is a hydroxybenzoic acid, a major component of vanilla flavor, which is used in the food industry for flavoring and preserving applications, along with the beverage, pharmaceutical, cosmetic, and tobacco industries [88].

Another predominant phenolic acid found in cereal bran is p-coumaric acid, which is the most available form of cinnamic acid and the most common cinnamic acid isomer in nature [89]. Coumaric acid, like the vast majority of phenolic acids, is found in lower concentrations in the endosperm of grains, with significantly increased concentrations in the peripheral tissues. Barley (230 μg/g), wheat (166 μg/g), oats (165 μg/g), and corn (2555 μg/g) have the highest levels of p-coumaric acid in their pericarp fractions [89].

Generally, phenolic acids have powerful biological effects such as antioxidant activity, antibacterial activity, anticancer properties, and anti-inflammatory effects, along with flavonoids. Flavonoids derived from grain-processing by-products contain antioxidant and anti-inflammatory functional groups. Flavonoid glycosides, such as cyanidin 3-glucoside, have been found to provide antioxidant and anti-inflammatory effects, and their gastrointestinal absorption rate is high. These chemicals’ bioavailability is poor, ranging from 1–100 nmol/L of the total plasma concentration [87]. However, bounded polyphenols can be released in the colon by intestinal bacteria and certain enzymes, resulting in bioavailable phenolic metabolites with possible health advantages.

A study conducted on the effects of phenolic compounds on the large intestine’s modulation showed that chlorogenic acid was converted into caffeic acid during the first stage of microbial biotransformation (dehydroxylation, dehydrogenation, or ester hydrolysis), and the primary metabolites found included di- and mono-hydroxylated phenylpropionic acids, m-coumaric, and hippuric acid [90].

An in vivo study conducted by Nuria Mateo Anson and colleagues [91] investigated how processed bran from whole wheat bread modifies the bioavailability of phenolic acids and exerts antioxidant and anti-inflammatory effects by examining postprandial plasma. The results showed a two- to three-fold increase in the bioavailability of ferulic acid, vanillic acid, sinapic acid, and 3,4-dimethoxybenzoic acid after eating bioprocessed bread. Furthermore, the ratio of pro-anti-inflammatory cytokines was significantly lower [92]. Another study examined the impact of regulated ferulic acid intake on lipid profiles, oxidative stress, and inflammatory activity in hyperlipidemic individuals. The improvement of the lipid profile, oxidation of LDL-cholesterol, decrease in oxidative stress, and reduction in inflammation were noticed after the consumption of 1000 mg of ferulic acid daily [93]. Thus, a diet rich in phenolic compounds can help prevent illnesses caused by oxidative stress and inflammation, as well as relieve symptoms associated with chronic diseases, including cardiovascular disease, obesity, and metabolic syndrome.

## 4. Polyphenols in Tomato-Processing By-Products

Tomatoes are highly appreciated and consumed for their appetizing taste and various shapes and colors, as well as for their antioxidant properties and chemoprotective and cardioprotective effects. Tomato-based food products such as juices, ketchup, sauces, paste, and puree generate large quantities of wet solid by-products with harmful environmental consequences; however, paradoxically, these solid wastes are a promising source of health-promoting compounds, such as carotenoids and polyphenols [2,22,94,95]. The main tomato-processing by-products are peels and seeds, together with small amounts of pulp. For instance, the seeds account for 10% of the fruit and 60% of the total waste [96].

Tomatoes contain various phytochemicals and bioactive compounds that are able to endure industrial treatments and still remain in the tomato waste [97]. Therefore, the valorization of tomato by-products could be a valuable source of natural bioactive molecules for human health. Their reintegration into the food chain may be achieved by using them as natural additives in food production and/or nutraceuticals [2]. Bioactive compounds found in tomato by-products are carotenoids, polyphenols, vitamins, proteins, and high-quality fatty acids [95,98,99]. Among these active molecules, polyphenols have antioxidant and anti-inflammatory properties that play an essential role in human health [25]. The primary phenolic compounds found in tomato by-products are flavonols and phenolic acids, illustrated in Figure 5, with quercetin, naringenin, and rutin being the predominant molecules [100].

Tomato peels contain a significantly higher level of phenolic compounds in comparison with the seeds and pulp; 83% of the flavonols in tomatoes are present in the peels [101]. As an example, a higher level of polyphenol content in tomato peel by-products (33.5 mg TAE/100 g dried peel) was reported by Sarkar and Kaul (2014) compared with tomato seed by-products (20.11 TAE/100 g meal) [94]. However, the profile of polyphenol contents found in tomato by-products is dependent on several factors, such as the fruit varieties [99], the geographical origin of the fruits [101], and the extraction methods used on the target compound [102]. Some examples regarding the total phenolic content in tomato by-products considering location and extraction method, as well as their antimicrobial and antioxidant activity, are provided in Table 3. Given the fact that tomato-processing by-products are most abundant in carotenoids as primary bioactive molecules, the antioxidant activity and antimicrobial effects might be attributed to the synergistic antioxidative effects of lycopene with other bioactive compounds to enhance their overall antioxidant activity [103].

The antioxidant and antimicrobial capacity of polyphenols has attracted increasing interest. These compounds have the ability to scavenge free radicals and reactive oxygen species, which are known to be involved in the development of cardiovascular diseases and several cancers [106]. For a more thorough study regarding the mechanism of action of polyphenols, Zhang and Tsao (2016) critically reviewed their biological activities, including antioxidative stress activity, anti-inflammatory effects, etc. [107].

The antioxidant capacity of tomato by-products can be divided between the hydrophilic and the lipophilic fractions. The antioxidant activity generated by polyphenols is attributed to the hydrophilic fraction since these molecules are soluble in water. The entrapment of polyphenols in the major insoluble fraction of tomato peels, formed mainly by hemicelluloses, leads to low antioxidant activity. The antioxidant activity of tomato peel fiber was found to be 3.90 µmol TE/g for the hydrophilic extract, lower than expected [105]. The mean value of the antioxidant activity of ten varieties of tomato by-products was found to be about 4.90 µmol TE/g, with the highest value being 5.74 µmol TE/g. Furthermore, little to any correlation has been found between the total phenolic content and antioxidant activity [99]. However, antioxidant activity might not always correlate with the number of phenols; it could be connected to the mutual interactions between all hydrophilic antioxidants and other constituents of the tomato by-products [95].

Even though tomato peels and seeds have relatively modest antioxidant activities, it is necessary to consider other remarkable attributes of these by-products, such as the antimicrobial capacity of polyphenols. Due to their hydrophobicity, phenolic compounds such as flavonols are capable of penetrating the cell phospholipid membranes of bacteria, able to exert their antibacterial activity inside the cell [108]. Gram-negative bacteria have cell walls surrounded by external membranes, which have high lipopolysaccharide contents. This makes Gram-negative bacteria less sensitive to the action of bioactive compounds such as polyphenols. This phenomenon was proved when the phenolic extracts of ten varieties of tomato by-products were tested against seven bacterial strains. The Gram-positive bacteria showed a higher sensitivity to the extract compared with the Gram-negative bacteria. All extracts were effective against *Staphylococcus aureus*, especially the Tiny Tim tomato variety, with a higher amount of flavonol glycosides and isochlorogenic acid. The Mirsini extract presented moderate antibacterial activity against Gram-negative bacteria (*Pseudomonas aeruginosa*, *Salmonella typhimurium*, *Klebsiella pneumoniae*, and *Escherichia coli*) [99].

The microbial contamination of soil can also influence the antimicrobial activity of fresh tomato by-products. When the Portuguese national microbiological guidelines for ready-to-eat foods were considered, unwashed and impure tomato by-products showed unsatisfactory *Enterobacteriaceae,* yeast, and *Bacillus cereus* spp. microbial counts for these food products. Washed and disinfected tomato by-products maintained these bacteria within satisfactory limits. Thus, tomato by-products have antimicrobial potential and can be used in food formulations or for cooking also considering their carotenoid content as natural food colorants. Furthermore, 6 months of frozen storage causes no substantial variation in microbial counts. Storing dry tomato by-products at room temperature slightly decreases the microbial counts of tested microorganisms [109]. Therefore, tomato by-products may control microbiological growth and may play an important role in the prevention of foodborne diseases.

## 5. Health-Promoting Properties of Polyphenol-Rich Diets

Since ancient times, the consumption of fresh vegetables, various green leaves, seeds, and fruits has promoted an optimal health state in the human organism without people knowing the exact mechanism of action. Polyphenols have contributed significantly to the enhancement of health status in humans over time, mainly due to their antioxidant properties, prebiotic potential, and, in some cases, their antimicrobial effects [25,110]. Based on their antioxidant effects, the polyphenolic structures are involved in the stimulation of the immune response against the progression of degenerative disorders such as metabolic syndrome, neurological affections, and different types of cancer. In addition, diets rich in polyphenols have been proven to sustain the optimal functioning of the immune system, as they enhance the activation of the NF-κB signaling pathway in macrophages that intervene in the management of inflammation processes [111]. Furthermore, multiple associations have been made between polyphenols and protective effects against chronic diseases based on chronic inflammation, and substantial evidence highlights the precautionary consequences of a polyphenol-rich dietary pattern [112,113,114].

The generous number of scientific reports available in recent years point out the strong antioxidant potential of vegetable-based polyphenols manifested through their antidiabetic, anti-obesity, anti-inflammatory, anticancer, hepatoprotective, cardioprotective, nephroprotective, neuroprotective, and osteoprotective effects, confirmed through various in vitro and in vivo studies [112]. In the same line, Condezo-Hoyos et al. highlighted, through a systematic review, the role of polyphenol-rich diets in clinical trials. The main results of the investigation showed substantial differences considering the consumption of 2564 mg of polyphenols (including hydroxycinnamic acids) per day [115].

Apples are among the most polyphenol-abundant fruits that are broadly consumed worldwide. Both the peels and pulp of apple fruits are rich in polyphenols, such as catechins, quercetin, rutin, phloridzin, phloretin, and chlorogenic acid, which are confirmed to exert positive effects, especially in reducing the evolution of neurodegenerative disorders [31]. At the same time, in vitro studies conducted on mice have proved that polyphenols extracted from apples have a protective effect on damaged gastrointestinal mucosa induced by drugs [31]. In addition, plenty of in vitro studies and clinical trials demonstrate the positive biological effects of polyphenol-rich foods such as apples on human and animal health, especially by stimulating the immune system. Diets consisting of moderate apple consumption maintain the general health state and accelerate the immune system’s response to pathogenic threats [116]. Due to the lower water percentage in apple by-products, the phenolic content increases compared with fresh fruit. Additionally, the exterior layer of the apple contains enzymatic and nonenzymatic antioxidants as a consequence of outer environment exposure, which is valuable for homeostasis maintenance [117].

In addition, cereal bran represents a valuable source of polyphenolic compounds. Bran derived from wheat, corn, rice, barley, sorghum, rye, oat, and millet has been proven to contain high concentrations of phenolics, flavonoids, and anthocyanins, which can inhibit the oxidation of human low-density lipoprotein (LDL) cholesterol. Moreover, the bioactive compounds found in cereal bran have great biological potential in the prevention of chronic disease development, such as cardiovascular disease, diabetes, and cancer [118].

In the same line, tomato-processing by-products contain different concentrations of polyphenolic compounds, mainly phenolic acids and flavonoids, which have been validated to be involved in decreasing triglyceride levels and regulating lipid metabolism. Nonetheless, tomato peel-derived polyphenols have shown anticarcinogenic, anti-inflammatory, anti-aggregation, and vasodilating effects in vitro and in clinical studies [119,120].

### 5.1. The Prebiotic Potential of Phenolic Compounds

Phenolic compounds may play an important role in the gut microbiota’s health state, acting as a prebiotic substrate for a diversity of probiotic strains [25,121]. In addition, polyphenols are considered prebiotics in the consensus document of the International Scientific Association for Probiotics and Prebiotics (ISAPP) [122]. Polyphenols such as catechin, gallic, vanillic, ferulic, and protocatechuic acids selectively stimulate probiotic strains and inhibit pathogenic ones. A study conducted by Pacheco-Ordaz et al. [123] investigated the effect of the five abovementioned polyphenols on the development of two probiotic strains, *Lactobacillus rhamnosus* GG ATCC 53103 and *Lactobacillus acidophilus* NRRLB 4495, and two pathogens, *Escherichia coli* 0157:H7 ATCC 43890 and *Salmonella enterica* serovar Typhimurium ATCC 14028. The authors observed that the combination of phenols allows for the selective growth of probiotics and inhibits the development of pathogenic bacteria [123].

Apple pomace, composed especially of seeds and peel, contains the more significant part of phenolic compounds, primarily phloridzin, chlorogenic acid, and soluble/insoluble fibers. These contribute to the diminished glycemic index and amended mineral bioavailability [124]. Besides improving the nutritional quality of food products, their applicability in food fortification promotes the viability and fermentation period of lactic acid bacteria, which can further upgrade sourdough quality [3].

### 5.2. Cardioprotective Effects of Polyphenols

The imbalance between free radical accumulation and antioxidant activity is one of the critical indicators contributing to the high prevalence of cardiovascular disease. This noncommunicable illness is the primary cause of mortality worldwide [125]. The current pandemic also highlighted the importance of nutrition and a healthy lifestyle regarding the immune system and the prevention of certain chronic diseases, particularly cardiovascular disease [126].

Diets comprising polyphenols extracted from apples, cereals’ bran, tomato peels showed positive activity close-connected to the well-functioning of the cardiovascular system [31]. For instance, in vivo studies on mice with cardiovascular disorders, treated with polyphenolic extracts from apple peels, showed improved results considering blood pressure, endothelial function, lipid homeostasis, and insulin resistance compared with the control group [127].

A double-blind, placebo-controlled investigation on 24 hyperlipidemic adults found that taking 1000 mg of ferulic acid daily for six weeks resulted in a significant reduction in total cholesterol (8.1%), LDL-cholesterol (9.3%), triglyceride (12.1%), and significantly increased HDL-cholesterol (4.3%) compared with placebo subjects [73]. Additionally, studies have demonstrated that regular whole grain consumption lowers the chance of developing particular illnesses, including type 2 diabetes, cardiovascular diseases, metabolic syndrome, and obesity [72,103,104]. Particular attention has been paid to the use of polyphenols in the prevention of cardiovascular-associated diseases, as these represent one of the main causes of increased mortality worldwide [105]. Moreover, ferulic acid, which abounds in wheat and oat brans, has good cardioprotective potential, as it maintains blood pressure, cholesterol, and glycemia at their optimal levels [128]. Furthermore, phenolic compounds found in tomato peels and seeds influence the cardiovascular system, mainly through their antithrombotic activity, imprinting ischemic injury prevention at the same time [129].

### 5.3. Polyphenols in Weight-Control Diets

Gut homeostasis is perturbed by obesity and overweightness, and it consequently enhances the ratio of *Firmicutes*/*Bacteriodetes*, which accelerate adiposity [130]. Studies conducted on polyphenols extracted from apples showed the positive impact of these compounds on weight loss and on the microbial communities of the guts of obese individuals [131]. Moreover, studies conducted on the effect of phenolic acids extracted from grain bran showed their output in reducing obesity risk and attenuating insulin resistance, thus contributing to maintaining a balanced body weight [132]. Polysaccharides, such as arabinoxylans and xylooligosaccharides extracted from cereal by-products, have a prebiotic effect, which also contributes to various metabolic dysfunctions and decreases blood cholesterol and glucose [133,134].

Polyphenols have shown a positive impact on the control of blood glucose levels, thus resulting in their positive attributes in the management of metabolic-related afflictions, such as diabetes [135]. In a systematic review performed by Coe and Ryan, several randomized clinical trials studying the effect of polyphenols in connection with carbohydrates on acute postprandial glycemic and insulin responses were analyzed. The results of the study pointed out that polyphenol-rich sources diminished the peak and early-phase glycemic response and also maintained the glycemic response in the later digestion phases. In addition, the polyphenol intake exhibited the mitigation of the insulin response, in particular, when they were consumed with bread, thus sustaining the beneficial properties in diminishing type 2 diabetes incidence [135].

## 6. Applicability of Recovered Polyphenols from Apple-, Cereal-, and Tomato-Processing By-Products in Functional Food Products in the Food Chain

The present concept of functional foods is the result of the gradual recognition that healthy diets derive from eating nutritious foods and from the identification of the mechanisms by which foods modulate metabolism and health. Some groups of foods, in addition to their nutritional properties, have supplementary properties for health. These types of foods are called functional foods and may be defined as any food that has a positive impact on an individual’s health, physical performance, or state of mind, in addition to its nutritious value.

### 6.1. Apple-Processing-Derived By-Products

The by-products of apple processing can be efficiently integrated into various foodstuffs, as specified in Table 4. Due to high fiber content, such as pectin and starch, which increases the viscosity of the products, these by-products can be integrated efficiently into functional food products, such as jellies, jams, or other foodstuffs [30]. In addition, due to several favorable health-related effects, the integration of AP is becoming more and more popular in various forms and applications [3,136].

One of the most efficient methods of preserving and integrating AP is drying and grinding it to obtain a powder, namely, AP flour. As indicated by Martau et al., 2021, the fortification of sourdough with 5 or 10% AP flour improved the microorganism’s viability due to high sugar content and enhanced the flavor of the sourdough [137]. Another study indicated an increased dietary fiber content after the inclusion of AP flour in wheat and rye crispbreads, from 9.39 to 15.89 g/100g and 15.8 to 19.89 g/100, respectively [138]. Besides, no browning reaction could be observed in the final products even if the AP flour contained increased reducing sugar content. The same results could be observed in other studies, as shown in Table 4.

AP can also be applied to snacks or yogurts through extrusion and freeze-drying or even to pork meat to prepare salami products [139,140,141]. The main phenolic compound, phlorizin, has also been efficiently integrated into chewing gums, which can be dissolved after 5 min of chewing, thus contributing to administering bioactive compounds [142].

On another note, from apple by-products, several other important compounds can be obtained, such as pectin, via extraction and various organic chemicals (e.g., butanol, propionic acid, bio-succinic acid) using fermentation processes [143]. In addition, from the extracted pectin, efficient biofilm solutions can be produced, which can be further applied in the production of active packaging that can be additionally enriched with AP-derived antioxidants [144]. Introducing these bioactive compounds results in biodegradable, water-soluble packages that improve the quality and shelf-life of wrapped foodstuffs.

### 6.2. Cereal-Processing-Derived By-Products

By-products belonging to the cereal industry mainly include rice, corn, wheat, barley, and other cereals used in the production of bakery products and alcoholic beverages. Cereal grains comprise one of the primary assets of human food, and recently, their manufacturing has expanded; the by-products originating from these resources have also increased [28,145]. The direct derivatives of these cereals are bran, germ, husk, hulls, and brewery wastes (brewer’s spect disposal) [133]. These comprise diverse, major compounds with significant health effects [14,134].

The need for gluten-free diets, especially in humans with Celiac disease, has grown steadily, and finding products with enhanced nutritional properties is of foremost importance [146]. The integration of by-products can contribute to the improvement of gluten-free products. A recent study improved the nutritional content of rice muffins through the integration of sweet corncob flour. This increased the fiber and ferulic content of the final product [147]. Another efficient method is the integration of oat proteins, which are usually removed through processing and can be efficiently integrated into various functional food products [148,149].

Several other kinds of cereal, native to their respective countries, contain important functional and nutritional properties. One of these is quinoa, a pseudo-cereal that originates from the Andean region. This cereal can also be efficiently used in the production of gluten-free products, especially after sourdough fermentation [150]. Through fermentation, the overall characteristics of the final foodstuff obtained from cereal by-products are improved, through prolonged shelf life, improved texture, flavor, diminished antinutrients, and growth in phytochemicals [151,152].

### 6.3. Tomato-Processing-Derived By-Products

Tomatoes are consumed in high quantities worldwide; thus, the by-products originating from this vegetable are consistent. Various studies have handled the reutilization of this by-product, as it contains essential amounts of organic acids, sugars, antioxidants, fibers, vitamins, proteins, and oils that are crucial to the effective functioning of the human body [22,153].

The main compounds recovered from tomato-industry-derived by-products are carotenoids, which can be efficiently integrated into various functional food products. They contribute to the sensory properties and shelf life and increase the bioactive ingredients [97,154,155]. Several studies tackle the troubles that are encountered through their inclusion in food products, such as their hydrophobic nature. A recent study by Szabo et al., 2022, efficiently incorporated these carotenoids in microcapsules that improved bioaccessibility through in vitro digestion model [2]. In a recent study, tomato by-products were integrated with various vegetable oils [154]. These extracts decreased the viscosity in the cases of hempseed and grapeseed oils and increased the viscosity in the case of flaxseed oil. This variation in viscosity can be attributed to the fact that the enrichment increased the thermal motion of the oil molecules and decreased the intermolecular resilience. These oils can be easily integrated with functional foods and are an efficient delivery system for these carotenoid extracts.

A high portion of tomato by-products is composed of seeds (5–10%), from which tomato seed oil can be extracted. These seeds consist of proteins (34%), lipids (30%), and fibers [129], as well as unsaturated fatty acids (palmitic, oleic, and linoleic acid). Generally, the proportion of oil extraction from tomato seeds ranges between 10 and 35% DW, depending on the extraction method [156].

**Table 4 molecules-27-07977-t004:** Examples of the reintegration of apple-, cereal-, and tomato-processing by-products in functional food products.

		Food Product	Effect On Food Product	Ref.
**Apple**	AP flour (5, 10%)	Bakery product—sourdough	+ cell viability; ↑ organic acid content (malic, oxalic, and citric acid)	[3]
	AP flour (5, 10, 15%)	Cereal crispbread	↑ total dietary fiber; ↑ hardness and crispiness	[139]
	Apple peel powder	Muffin	↑ dietary fiber; ↑ bioactive compounds; + color and texture; + organoleptic characteristics (12%)	[157]
	AP flour (2.5, 5, 7, 10%)	Pasta	↓ carbohydrate content; ↑ fiber, protein, fat, and ash content; ↑ swelling index, cooking water absorption, and cooking loss; ↓ optimum cooking time; ↓ texture and structure of pasta (10%)	[32]
	AP (2, 3, 6, 9%)	Freeze-dried snacks	↑ AP; ↓ lightness coefficient; ↑ cutting force; ↑ organoleptic properties (2%); ↓ water activity	[158]
	AP 20%	GF corn snacks	↑x36 chlorogenic acid; ↑x4 cryptochlorogenic acid; ↑x6 catechin; ↑x3 procyanidin; ↑x8 epicatechin; ↑x25 phlorizin; ↑x3 total soluble and insoluble dietary fiber; ↑ organoleptic scores	[159]
	AP powder	Yogurt	↓ sensory profile; ↑ protein and fat content; ↑ rheological attributes	[141]
	Freeze-dried AP powder (0.5, 1%)	Set-type yogurt	↑ gelation pH; ↓ fermentation time (1%); firmer and consistent yogurt (cold storage); stable structure (0.5%); stabilizer and texturizer	[43]
	Dried AP (7, 14%)	Italian salami	↓ fat and calories; ↑ fiber and phenol content	[140]
	Defatted apple seed flour	Chewing gum	↑ phlorizin content (52–67% and 75–83% of the total phenolics)	[143]
**Cereal**	WB & BF	Bread	↑ dietary fibers content	[160]
			↑ alveograph profile	
			↑ volume of bread	
	SCC (10, 20, and 30%)	GF rice muffin	↑ dietary fibers and ferulic acid content; ↑ nutritional value; + height, color, and texture (20% SCC)	[147]
	BRF	Buns and muffins	↑ dietary fibers, iron, zinc, and calcium; ↑ antioxidant capacity and phytonutrient content; ↓ carbohydrates and sensory acceptability; moderate glycemic index and glycemic load; ↑ shelf life	[161]
	OPC & OPI	Yogurt	↑ nutritional benefits (OPI); ↑ product quality and sustainability (OPC); ↑ nutritional (OPC)	[149]
	BMG + PBD (1:1)	Cereal composite bar	+ essential minerals and fiber; ↑ sensorial evaluation; antifungal properties	[162]
	BSG	Yogurt	↑ viscosity and shear stress; ↓ fermentation time; maintained flow behavior and stability	[163]
	BRG+PFPF+WP	GF breakfast cereals	Average acceptance; + total, soluble, and insoluble dietary fiber; ↑ darkness, protein, and carbohydrate content; ↓ expansion and consumer acceptance	[164]
	PH	Gel-based foods	↑ textural and sensory characteristics; syneresis and fat loss during cooking avoidance; ↑ gelling properties	[165]
**Tomato**	CT	Hemp, flaxseed, grapeseed oil	↑ oil quality; ↑ viscosity (flaxseed oil); ↓ viscosity (hemp and grapeseed oil); intense color	[154]
	TBPP	biofilms	↑ aesthetic impact and coloring; ↓ transparency	[166]
	TBPP	biofilms	↑ physical properties (diameter, thickness, density, weight); ↑ antimicrobial effect; ↑ total phenolic content	[155]
	TPP	GF ready to cook snack	↑ fiber, mineral, and lycopene content; ↑ antioxidant activity; ↓ oil uptake	[167]
	TPF	Spreadable cheese	↑ spreadability; ↑ antioxidant activity and phenolic content; ↑ fibers	[168]
	TBP	Passata	↑ total dietary fiber; ↑ lycopene and polyphenols	[169]
	TPP (5, 10, 15, 20, 25%)	cookies	↓ lightness values; ↑ redness and yellowness; acceptable by consumers (5%)	[170]
	TPF (15%)	pasta	↑ carotenoids and dietary fiber; ↓ sensory scores for elasticity, odor, and firmness	[171]

Food manufacturing through 3D-printing technology is an innovative method of delivering personalized food, meeting our nutritional needs and expectations regarding taste, texture, color, and other aspects [172]. In addition, personalized, 3D-printed food products are promising in the prevention of different noncommunicable diseases due to the possibility of enhancing them with bioactive compounds (e.g., polyphenols, dietary fibers, proteins, etc.) [173].

As for future perspectives, bioactive compounds, such as polyphenols, vitamins, and/or proteins found in food processing by-products, could be easily integrated into functional foods by associating 3D-printed food with molecular gastronomy [174]. Alongside a facile integration of recovered polyphenols, the inner biological activities and nutritional profile can be enhanced, obtaining fortified nourishment [175]. In addition, delivering fortified foods via 3D printing is a sustainable approach to food-waste management, according to circular economy principles [174].

## 7. Conclusions

All things considered, apple-, cereal-, and tomato-processing by-products are a valuable source of phenolic compounds that can be safely reintegrated with the food chain within recommended parameters. Apple pomace possesses a significant number of beneficial effects for the human body, most of them offered by their antioxidant and anti-inflammatory properties. The three main compounds identified in apple pomace—phlorizin, chlorogenic acid, and epicatechin—can act upon metabolic diseases, being important pillars in the prevention and treatment of the ailment.

Studies demonstrate that regular whole grain consumption lowers the chance of developing noncommunicable diseases, the cause for these health effects being the synergy between the polyphenols and dietary fibers found in the outer layers of the grains, which are unfortunately discarded as by-products.

It is necessary to consider the remarkable antimicrobial capacity of polyphenols derived from tomato by-products, with a focus on flavonol glycosides and isochlorogenic acid against *Staphylococcus aureus.*

In addition to altering the composition of the gut microbiota, which is closely linked with health benefits, gut bacteria also metabolize polyphenols to create bioactive chemicals that have therapeutic effects. Polyphenols such as catechin, gallic, vanillic, ferulic, and protocatechuic acids selectively stimulate probiotic strains and inhibit pathogenic ones.

In light of this, polyphenols seem to be promising candidates for use in both functional food products and personalized nutrition (e.g., 3D-printed food). To fully take advantage of their outstanding qualities, various pharmacokinetic concerns, such as decreased intestinal absorption and bioavailability and quick metabolic alterations, should be considered.

## Figures and Tables

**Figure 1 molecules-27-07977-f001:**
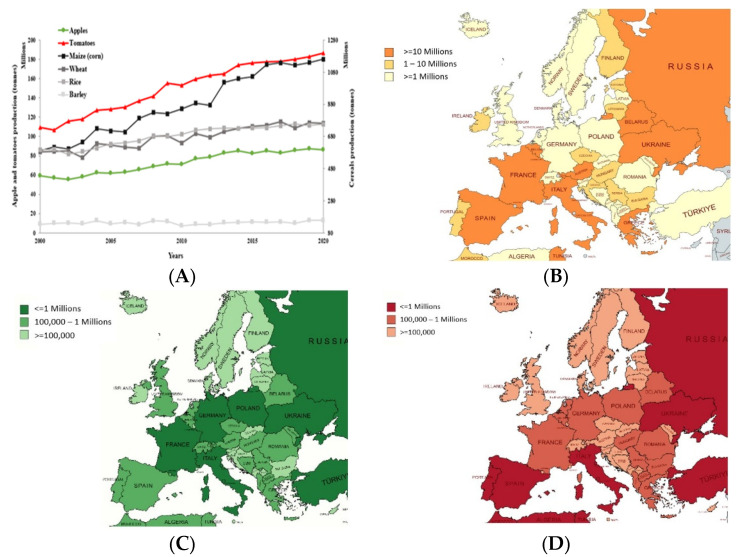
Production of cereal, apples, and tomatoes (tons) in Europe. (**A**) Evolution of world cereal, apple, and tomato production over the last two decades; (**B**) production of wheat (tons) in Europe for every country in 2020; (**C**) production of apples (tons) in Europe for every country in 2020; (**D**) production of tomatoes (tons) in Europe for every country in 2020 (http://www.fao.org/faostat, accessed on 22 April 2022).

**Figure 2 molecules-27-07977-f002:**
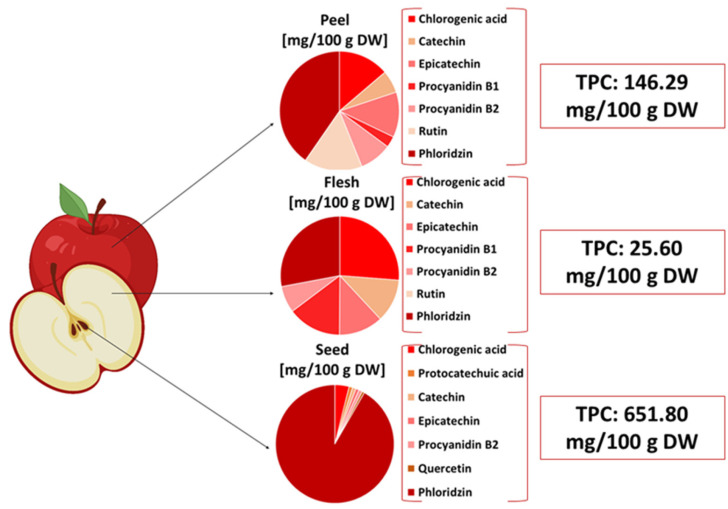
Distribution of phenolic compounds in apple fruit, according to Feng et al. [49]. TPC—total phenolic content; DW—dry weight.

**Figure 3 molecules-27-07977-f003:**
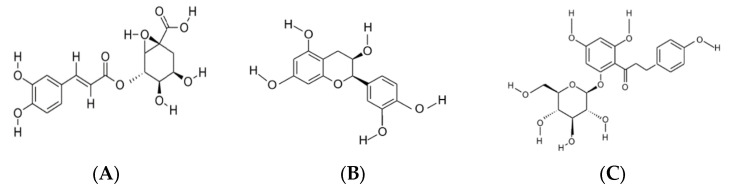
The chemical structure of the predominant phenolic compounds identified in apple pomace: chlorogenic acid (**A**); epicatechin (**B**); phlorizin (**C**). Source: ChemDraw Software.

**Figure 4 molecules-27-07977-f004:**
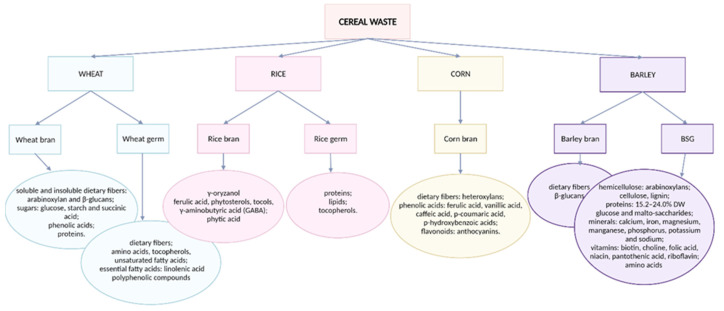
The most common cereals and their primary bioactive compounds.

**Figure 5 molecules-27-07977-f005:**
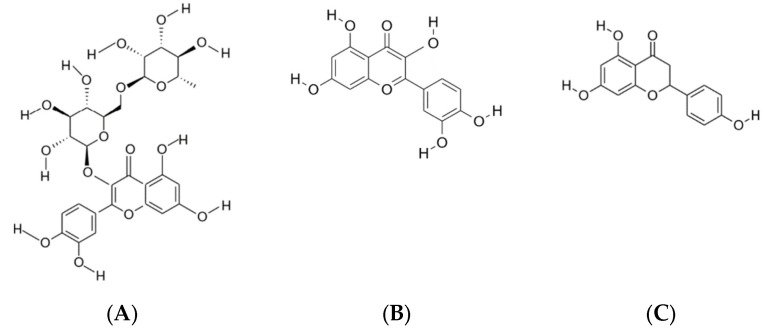
The chemical structure of the predominant phenolic compounds identified in tomato-processing by-products: rutin (**A**); quercetin (**B**); naringenin (**C**). Source: ChemDraw Software.

**Table 1 molecules-27-07977-t001:** The nutritional and polyphenolic profile of apple pomace ^1^ [43].

Composition	Amount (% DW)
Total sugar	45.1 ± 5.3
Total dietary fiber	26.5 ± 0.8
Insoluble fiber	18.4 ± 0.4
Soluble fiber	8.2 ± 0.5
Total phenolic content (mg EGA/100 g AP)	289.1 ± 24.2
Fat	3.8 ± 0.2
Protein ^2^	3.8 ± 0.0
**Polyphenolic profile**	**(mg/100 g dry matter)**
Quercetin-3-*O*-galactoside	22.55 ± 0.34
Quercetin-3-*O*-xyloside	13.91 ± 0.03
Quercetin-3-*O*-rhamnoside	19.21 ± 0.00
Chlorogenic acid	20.55 ± 0.12
p-coumaroylquinic acid	0.16 ± 0.03
Catechin	1.44 ± 0.02
Procyanidin B2	2.61 ± 0.00
Phloretin-2-*O*-xylosyl-glucoside	1.48 ± 0.14
Phlorizin	15.52 ± 0.00

^1^ Values represent mean ± standard deviation, based on [43]; ^2^ nitrogen to protein conversion factor was 5.7.

**Table 2 molecules-27-07977-t002:** Chemical composition of cereal bran.

Compound	Amount (% DW)				References
	Wheat Bran	Rice Bran	Oat Bran	Corn Bran	
Water	12.1	8.72-	29.4–31.2	4	[70,71,72,73]
Protein	13.2–18.4	10–16	5.9–6.7	9.5–10.1	[71,74,75]
Fat	3.5–3.9	15–22	6.47	1.92–6.41	[74,76,77,78]
Total carbohydrates	56.8	34.1–52.3	66.22	78.05–79.7	[74,76,77,78]
Starch	13.8–24.9	18.19–32.45	2.5–16.3	27.7–28.2	[70,72]
Cellulose	10.5–14.8	15.8	3.4	23–23.1	[72,79]
Hemicellulose	35.5–39.2	31.3	35%	26.1–27	[72,79,80]
Lignin	8.3–12.5	11.6	11.22	2.2–6.5	[72,79]
Total arabinoxylans	10.9–26.0	4.8–5.1	3	17.5–17.7	[70,79,81]
Total β-glucan	2.1–2.5	0.04–0.21	5.4–8.5	-	[79,81]
Phenolic acids	1.1	1.57	0.7–1	2.2-2.7	[71,74,82]
Ferulic acid	0.02–1.5	0.004	1.76	1.5–1.9	[70,82,83]
Phytic acid	4.2–5.4	50.68 *	-	-	[74]
Ash	3.4–8.1	10.65	10.3–10.9	4	[71,73]

* Phytic acid content of rice bran is expressed as mg/g DW.

**Table 3 molecules-27-07977-t003:** Total phenolic content and antimicrobial and antioxidant activity considering the localization and extraction methods.

Geographical Origin	Tomato By-Products	Extraction Method	Total Phenolic Content	Antioxidant Activity	Antimicrobial Activity	Reference
India	Peels	Solvent extraction	33.5 mg TAE/100 g	21.0% inhibition/g	-	[94]
	Seeds		20.11 mg TAE/100 g meal	34.0% inhibition/g	-	
Romania	Seeds and peels of 10 varieties of tomato		111.9 to 407.7 mg/100 g DW	Mean value of 489.9 ± 41.5 µmol TE/100 g	Gram-positive and Gram-negative bacteria	[99]
	Peels of 10 varieties of tomato		35 to 157 mg/ 100 g DW	Mean value of 201 ± 44 µmol TE/100 g		[95]
Portugal	Whole tomato		408.89 ± 12.11 and 277.24 ± 11.29 mg GAE/100 g DM (fresh and after 6 months of frozen storage); 310.33 ± 10.38 and 283.64 ± 11.84 mg GAE/100 g DW (before and after 6 months of powder storage)	ABTS (694.07 ± 45.00 and 558.73 ± 29.06 mg TE/100 g in fresh and after 6 months of frozen storage; 350.15 ± 14.37 and 407.56 ± 25.93 mg TE/100 g before and after 6 months of powder storage)	*Enterobacteriaceae*, *Bacillus cereus* spp., yeasts, and molds	[104]
				ORAC (3165.18 ± 77.48 mg TE/100 g, 3285.77 ± 271.25 mg TE/100 g 1771.66 ± 31.25 mg TE/100 g in fresh and after 1 to 6 months of frozen storage; 1581.76 ± 124.90 TE/100 g, 1610.74 ± 46.51 mg TE/100 g and 1229.74 ± 38.52 mg TE/100 g before and after 2 to 6 months of powder storage)		
				DPPH (418.79 ± 30.92, 648.06 ± 55.38, 388.53 ± 27.18 mg TE/100 g in fresh and after 3 to 6 months of frozen storage; 117.78 ± 4.99 to 130.44 ± 3.51 mg TE/ 100 g before and after 6 months of powder storage)		
Spain	Peels fiber	Enzyme hydrolysis	291.14 ± 11.1 to 353.15 ± 19.6 mg GAE/kg	3.90 µmol TEAC/g	-	[105]
		Maceration	749.84 ± 15.55 mg GAE/kg			
		Ultrasonic assistance (5 to 15 min)	985.78 ± 112.93 to 1056.18 ±67.9 mg GAE/kg			

DW—dry weight; GAE—gallic acid equivalents.

## Data Availability

Not applicable.

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
