# Peer review of "Natural Polyphenol Recovery from Apple-, Cereal-, and Tomato-Processing By-Products and Related Health-Promoting Properties"

_molecules, 2022, doi:10.3390/molecules27227977_

Round 1
Reviewer 1 Report
The review Natural polyphenols recovery from apple, cereal, and tomato processing by-products, and related health-promoting properties is an interesting nowadays topic. The authors presented three main products as apple, cereals and tomatoes and highlighted the phenolic aspect and health related properties. Interesting is the potential application of by-products in foodstuff. However, major revision are needed.
-Figure 1. Only the first Figure (fig 1A, suppose) is mentioned, while the European situation is not analysed. Worldwide or globe situation is considered. I think that should be defined a geographical origin of the data or start from worldwide and focus on Europe in order to present the other three figure 1 (b,c,d).
- line 117-19 seems that the statement is unlinked with the previous one because describe only grains. Moreover, the aim is presented immediately after.
-lines 128-135. There is a non linear transitin between worldwide and Europe. Should be better present the wolrd situation and then focuse on Europe due to the presence of Figure 1(b,c,d) related to Europe
-149-151 it is a repetition of nutritional compounds already listened in line 147. The information can be added in line 147-149 where the positive effects are listened.
-line 152. The nutritional profile of apple pomace is not of interest, what is the aim is the phenolic profile.
-line 152: space 1[39]
-Lines 155-162: The toxicity is associated to seeds. If the the apple pomace is not used as such for the human consumption, the problem still remani? Is it known the percentage of seeds in AP? Technological processings (extraction, thermal treatments and others can modified the toxin lowering the effects?
Probably these considerations should be provided.
-line 177: Check the correspondence with the Figure 2A
-line 187: delate "is" in "due to its ability to alter the glucose is absorbed and excreted"
-line 195:Check the correspondence with the Figure 2B
-line 207: Check the correspondence with the Figure 2C
-line 236: Table title: Is it correct use "phytochemicals" if the table reports also the composition, also with macromoleculs, of the bran? Probbaly is better change the title of the table as "Chemical composition....", mainly because the section refers to "phenolic compounds...."
-286-288: Already mentioned in lines 87-89
- Section 4. Polyphenols in Tomato processing by-products. It is completely different form the form and presentation of the other two. The table and info related to extraction process, and antioxidant and antimicrobial activity, barely mentioned in the sections 2 and 3. I think that a uniform presentation should be considered.
- The health related properties stated for polyphenols in general, few data regarding the by-products object of this review. The health related properties of polyphenols are well know. I think that should be linked well with by-product or at least recalled in the text.
- in section 5.3. Polyphenols in weight control diets, use beans is not the right examples. If no studies are reported linked to polyphenols recorded in tomato, apple and cereal by products, delete the section.
- line 486: space "Reintegrationof"
- line 492, 519, 542: identified the sections, use italics?
- section 6.2. Future perspectives regarding personalized nutrition (e. g. 3D food printing): I think that this section is forced and extra with respect to the aim of the study. Some statements could be inserted in food application.
Author Response
We would like to kindly express our appreciation for your time and effort in reviewing the manuscript. The suggested modifications were helpful to improve the quality and the content of the paper, therefore we agreed and applied the comments and recommendations as described in the point-by-point response in the attached document.

Reviewer 2 Report
Manuscript iD: molecules-1999783
Authors: Szabo et al.,
In this review article entitled “Natural polyphenols recovery from apple, cereal, and tomato processing by-products, and related health-promoting properties”, the authors studied the processing-derived by-products, specifically polyphenols from apple, cereal and tomato and summarized their beneficial effects on human health.
Hereafter, some points that should be taken into account before processing further.
Comments to the authors:
- The text within the figure 2 is not clear. The authors are requested to make it clearer and increase the font size.
- Regarding the 3D chemical structure of the compounds, it would be better to use an appropriate software such as ChemDraw or Accelrys to draw the structure of these polyphenols.
- More attention should be given in “–“ and “-“ (dash and minus, respectively), specifically in table 2.
- The references, which are given in table 2 should be inserted between brackets as elsewhere.
- English language is fine. Just small checking/editing is required.
- What are the reasons behind the selection of wheat only, as cereal type, for the production (in tones) in Europe.
- More clarification is needed for tomato CT and how it increased and decreased the viscosity at the same time as an effect on food product?
Author Response
We would like to kindly express our appreciation for your time and effort in reviewing the manuscript. The suggested modifications were helpful to improve the quality and the content of the paper, therefore we agreed and applied the comments and recommendations as described in the attached document.
Thank you.

Round 2
Reviewer 1 Report
Comments are fully addressed.
Reviewer 2 Report
Manuscript : molecules-1999783 (Round #2)
Authors: Szabo et al.,
Title : Natural polyphenols recovery from apple, cereal, and tomato processing by-products, and related health-promoting properties
After providing a revised version along with a response to reviewer’s comments, we can obviously notice that the manuscript has been improved but somehow my comment No. 2 was overlooked. Hence, I guess a minor revision is still needed.
- If the authors want to keep the chemical structures, which are depicted from pubchem, then it would be better to provide the 2D structures as they are much better in publication quality. 3D structures are good if we have the possibility to change their orientations, which is impossible in the publications.
- Still more attention is required for “-” and “–“ (dash and minus), specifically in table 2.
Author Response
We would like to kindly express our appreciation for your time and effort in reviewing the revised version of the manuscript. We implemented the comments and recommendations as described in the point-by-point response below:
|
Comment 1: If the authors want to keep the chemical structures, which are depicted from pubchem, then it would be better to provide the 2D structures as they are much better in publication quality. 3D structures are good if we have the possibility to change their orientations, which is impossible in the publications. |
Response to comment 1: Thank you very much for your suggestion. We decided to construct the chemical structures of the interest compounds by ChemDraw Software, accordingly to your previous recommendation. Please find the 2D structures in the revised version of the manuscript. |
|
Comment 2: Still more attention is required for “-” and “–“ (dash and minus), specifically in table 2 |
Response to comment 2: Thank you for the meticulous observation, we made the required corrections in the text, specifically in table 2. |